# A SYSTEMATIC COMPARISON OF SYLLOGISTIC REASONING IN HUMANS AND LANGUAGE MODELS

## ABSTRACT

A central component of rational behavior is logical inference: the process of determining which conclusions follow from a set of premises. Psychologists have documented several ways in which humans' inferences deviate from the rules of logic. Do language models, which are trained on text generated by humans, replicate these biases, or are they able to overcome them? Focusing on the case of syllogisms—inferences from two simple premises, which have been studied extensively in psychology—we show that larger models are more logical than smaller ones, and also more logical than humans. At the same time, even the largest models make systematic errors, some of which mirror human reasoning biases such as ordering effects and logical fallacies. Overall, we find that language models mimic the human biases included in their training data, but are able to overcome them in some cases.

## 1 INTRODUCTION

The capacity to reason deductively—that is, to determine which inferences, if any, follow from a given set of premises—is central to rational thought (Newell & Simon, 1972; Laird et al., 1987; Fodor & Pylyshyn, 1988; Griffiths et al., 2010). Still, human reasoning often displays systematic biases (Gigerenzer & Gaissmaier, 2011; Marcus, 2009; Kahneman, 2013; McClelland et al., 2010). In recent years, neural network language models (LMs) trained using self-supervised objectives have been reported to display a range of capabilities, including the ability to reason (Brown et al., 2020; Chowdhery et al., 2022; Bubeck et al., 2023). Do LMs' logical reasoning abilities follow the rules of logic to a greater extent than humans'? To the extent that LMs' reasoning deviates from normative logic, are their biases similar to humans biases (Binz & Schulz, 2023; Dasgupta et al., 2022)?

In this work, we address these questions with a detailed study of a particularly simple case— inferences from pairs of premises, or *syllogisms*:

> **No artists are bakers**,
> and **All bakers are chemists**,
> - - - - - - - - - - - - - - - - - - - - - - - - - - - - - -
> therefore: **some chemists are not artists**.

In a syllogism, each premise relates two terms with one of four quantifiers (or "moods": *all*, *some*, *none* and *some are not*) and only one term is shared between the premises (here *bakers*), and so deducing a relationship between the terms not shared (*artists* and *chemists*) requires inference.

Humans show a wide range of behaviours when making syllogistic inferences, often deviating from logic; in fact, for some syllogisms the vast majority of participants draw incorrect inferences (Khemlani & Johnson-Laird, 2012). This could pose a challenge to language models (LMs), which learn from corpora consisting primarily of human-generated text, which reflects human beliefs and inferences. Is there sufficient signal in the training corpus to steer LMs away from the (often incorrect) human inferences and toward normative logic, whic is desired for most applications? We perform a detailed comparison of the performance of the PaLM 2 family of transformer LMs (Google, 2023) with findings from the human syllogistic reasoning literature.

We report the following results:

1. LMs draw correct syllogistic inferences more often than humans and larger LMs are more accurate than smaller ones, but even the largest LM obtains an accuracy of only about 80% (Section 4.1).

2. LM errors are systematic, with very low accuracy on particular syllogism types (Section 4.1); the syllogisms that LMs struggle with are a subset of those that humans find difficult (Section 4.2).

3. Like humans, LMs are sensitive to the ordering of terms in the premises of a syllogism even when it is logically irrelevant (Section 4.2; this pattern is known as the "figural effect" in cognitive psychology; Johnson-Laird & Steedman 1978).

4. LMs show many of the same *syllogistic fallacies* (characterized by high confidence and low accuracy) as humans. The largest LM is more susceptible to these fallacies (Section 4.2; Khemlani & Johnson-Laird 2017).

5. We use the Mental Models theory to show larger LMs show signatures of being more deliberative in reasoning, irrespective of their accuracy on the syllogisms (Section 5; Khemlani & Johnson-Laird 2022).

Overall, we find that LMs replicate many of the human biases discovered in psychology studies, consistent with the fact that LMs are trained on human-generated text. At the same time, for some syllogisms, sufficiently large models overcome those biases and achieve dramatically better accuracy than humans.

## 2 BACKGROUND

### 2.1 SYLLOGISMS

Syllogisms are logical arguments consisting of two *premises* relating three variables, A, B and C (e.g., artists, bakers, and chemists in the previous example). Each premise relates just two of the variables. The variables can be related by one of four quantificational statements, often referred to as "moods" in the classic literature (Table 1, left). The variables in the premises can be ordered in either of the two directions—e.g., *all artists are bakers* vs. *all bakers are artists*—and so there are four possible pairs of orderings (Table 1, right). These orderings are traditionally refered to as numbered "figures" (e.g. figure 1), but we will use the less confusing term "variable ordering". Taking the crossproduct of these building blocks yields 64 possible syllogisms: two premises, each of which can take one of four quantifiers and one of two possible orderings.

Though the premises only relate A and B, or B and C—never A and C—27 of the 64 syllogisms imply a quantified relationship between A and C (e.g., *some A are C*). In the remaining 37 syllogisms, no relation between A and C can be deduced; in human experiments, the expected response to these syllogisms is "nothing follows".

### 2.2 HUMAN SYLLOGISTIC REASONING

Cognitive psychologists, going back to the early 20th century, have found that in many cases the conclusions that humans draw from the premises of a syllogism deviate from logical norms (for a review, see Khemlani & Johnson-Laird 2012). These errors are systematic: some syllogisms are much harder than others, and the incorrect conclusions that participants tend to draw are consistent across participants. For example, the vast majority of participants incorrectly conclude *no artists are chemists* in response to the syllogism in the beginning of Section 1 (we analyse similar cases in detail in Section 4.2).

Several other human reasoning biases have been documented. For example, when given a syllogistic argument where variables are ordered according to the variable ordering (A-B, B-C; variable

| | | | | | 1 | 2 | 3 | 4 |
|---|---|---|---|---|---|---|---|---|
| **A**: | **All** artists are bakers | **I**: | **Some** artists are bakers | | A-B | B-A | A-B | B-A |
| **E**: | **No** artists are bakers | **O**: | **Some** artists **are not** bakers | | B-C | C-B | C-B | B-C |

Table 1: Syllogism moods (left) and variable orderings (right).

ordering 1), participants show a pronounced bias toward predicting conclusions with a A-C variable ordering, even though the variable ordering in the premises is irrelevent to a syllogism's logical content: reordering the premises does not effect the conclusions it implies (Johnson-Laird & Steedman, 1978). Participants are also likely to produce a conclusion when it is true in the real world, independently of whether it follows from the premises ("content effects", Evans et al. 1983).

Several theories have been proposed to explain human syllogistic reasoning. An influential account that we focus on in the present work is the Mental Models Theory (Johnson-Laird & Byrne, 1991). This theory posits that human reasoners construct "mental models" populated by a small number of entities that instantiate the premises; e.g., to instantiate *all artists are bakers*, a reasoner might construct a world in which there are three specific artists, all of whom are bakers. These worlds are constructed based on a number of fallible heuristics, and human reasoning errors arise when those heuristics produce incorrect conclusions (see Section 5).

### 2.3 LANGUAGE MODELS AND REASONING

LMs trained with self-supervised objectives on large text corpora have been instrumental in achieving high performance on a range of tasks. Some of the tasks that LMs have shown promise in have been referred to as reasoning tasks, including commonsense reasoning, natural language inference, or question answering (e.g., Chowdhery et al. 2022). In this work, we focus more specifically on deductive logical reasoning: drawing conclusions that *must*, rather than is likely to, be true given the premises, and where the inference is based only on the premises, and does not rely on world knowledge. Unlike work on datasets collected from textbooks or through crowdsourcing, we perform a well-controlled, analysis of a simple logical task that has received a lot of attention in cognitive science.

Several studies have benchmarked LMs on logical reasoning tasks (Han et al., 2022; BIG-bench collaboration, 2022; Wu et al., 2023a; Betz et al., 2020; Saparov & He, 2022; Saparov et al., 2023) and examined LM reasoning biases (Dasgupta et al., 2022; Razeghi et al., 2022; Wu et al., 2023b; McCoy et al., 2023). Saparov & He (2022) take a similarly controlled experimental approach to ours (see also Saparov et al. 2023), but they analyze LMs' performance on formal logic rather than problems phrased in natural language as we do, and do not compare their results to humans. The closest study to ours is Dasgupta et al. (2022), which demonstrates content effects in a number of logical reasoning domains, including syllogisms. We extend their approach to study other aspects of syllogistic reasoning.

## 3 METHODS

### 3.1 DATA

**Human data** We use the data from Ragni et al. (2019), an online experiment where 139 participants responded once to each of the syllogisms. In each trial, a participant was presented with a syllogism and was instructed to choose among nine options: the eight possible conclusions and "nothing follows". The experimental trials were preceded by a brief training phase where participants were familiarized with the task.

**Materials for LM evaluation.** To reduce the likelihood that the items we used have appeared in the models' training corpus, we generate our own dataset of syllogisms. To do this, we generate 30 content triples—i.e., nouns to fill in the abstract terms A, B, C in a syllogism—where there is no obvious semantic association between the terms, for example the triple "hunters, analysts, swimmers" (see Appendix A for the full list of content triples used). This procedure is similar to the one Ragni et al. (2019) used to generate their experimental materials.

### 3.2 MODELS AND INFERENCE

We evaluate the PaLM 2 family of LMs, which are publicly available in four sizes (XXS, XS, S and L; Google 2023). The PaLM 2 models are transformer-based (Vaswani et al., 2017) and were trained on a large corpus of multilingual web documents, books, code, mathematics, and conversational data.

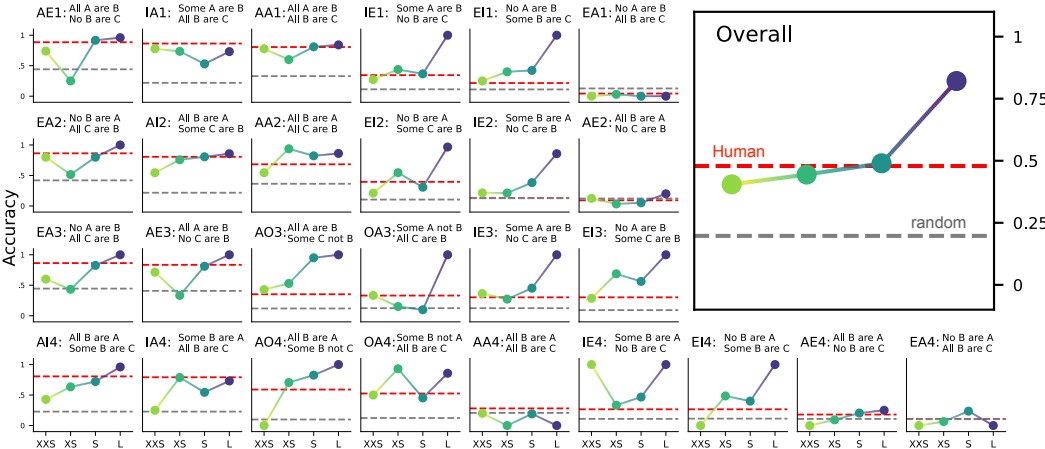

Figure 2: Accuracy of PaLM 2 models, humans (red), and random guessing (grey). Syllogisms are partitioned into variable ordering (by row) and ordered by decreasing human accuracy from left to right. The top right inset shows the average accuracy across all syllogisms. Syllogisms are identified with the letters of the moods and the premise and the number associated with their variable ordering.

Following the emerging standard practice for eliciting reasoning from large LMs, we use zero-shot "chain-of-thought" prompting (Kojima et al., 2022; Wei et al., 2022). We speculate that the more explicit reasoning process triggered by the chain-of-thought prompt may resemble the behavior of human participants in experiments more closely; for an analysis of alternative prompting strategies that we explored before settling on this one, see Appendix B.1. The prompt we use is illustrated in Figure 1. We randomize the order of the conclusions in the prompt to control for LMs' sensitivity to answer ordering (Pezeshkpour & Hruschka, 2023).

Following the prompt, we generate 75 tokens from the LM, with a temperature of 0.5. We repeat this process 30 times for each combination of syllogism type and content triple. We use uncased string matching to identify conclusions in the samples, filtering out samples for which no match was identified. We then take the conclusion that was produced most frequently across the 30 samples to be the model's answer on that syllogisms (see Appendix B for further details and an exploration of the impact of different prompts and decoding parameters).

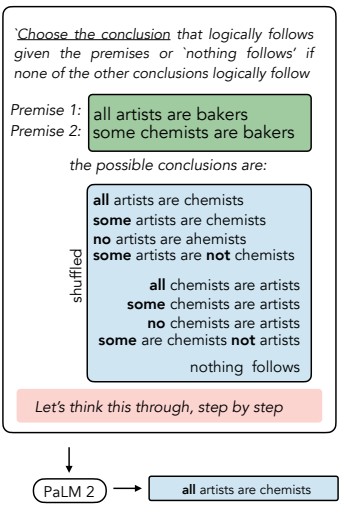

Figure 1: Example of the zero-shot chain-of-thought prompt we use to assess LM syllogistic reasoning.

## 4 RESULTS

### 4.1 DO LMS REASON ACCURATELY?

We first examine the LMs' behavior on each of the 64 syllogisms separately. The LMs rarely produced the output "nothing follows", which is the correct conclusion for 37 of the syllogisms. We return to this behavior briefly in Section 4.2, but in most of the following analyses we restrict ourselves to the 27 syllogisms that derive conclusion other than "nothing follows" (Figure 2). We compute the LMs' accuracy for each syllogism by dividing the number of logically valid conclusions produced by the LM by the total number of responses; note that some syllogisms have more than one valid conclusion (up to four) and so the random baseline in Figure 2 varies by syllogism.

When averaged across all syllogisms, LM accuracy improves with scale, with the largest model exceeding human performance. However, there is considerable by-syllogism variance; for multiple syllogisms, accuracy stays very low and can even decrease as model size increases (this is the case, for example, for *all B are A, all B are C*).

## 4.2 Do LMs Reason Like Humans?

Human accuracy averaged across all syllogisms roughly 50% (Figure 2; red-dashed line); as such, high LM accuracy on this task does not necessarily imply humanlike reasoning. We find that the syllogisms that models struggle with are ones that humans also find challenging, but the inverse is not always true: there are multiple syllogisms that are hard for humans but are solved correctly by larger models. For example, for the syllogism *some B are A, no B are C* human accuracy is barely above chance whereas PaLM 2 Large is near ceiling.

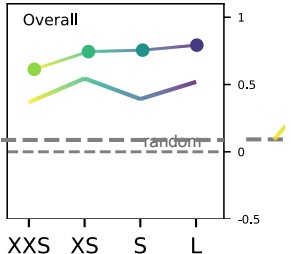

Figure 3: Correlation with human predictions for the PaLM 2 models.

**Comparing the distribution over responses.**   To compute the probability distribution over conclusions for each syllogism, we aggregate response counts for each syllogism and normalize them into a probability distribution as in Khemlani & Johnson-Laird (2016). Figure 3 shows the result of correlating the probability estimates from humans with the estimates from PaLM 2 models across the entire dataset (though see Figure 11 in the Appendix for a by-syllogism break down). Overall, larger models are more correlated with the human data than smaller ones (Figure 3). We note that PaLM 2 Large displays, at the same time, both a high correlation with human responses and a higher-then-human accuracy. This suggests that the miscalibration to human data that models accrue due to higher accuracy is balanced by better fit to humans elsewhere in the dataset. The next sections test this hypothesis, zooming in on two specific biases.

**Variable ordering effects**   Humans' syllogistic inferences are sensitive to variable ordering, even when it is logically irrelevant (Johnson-Laird & Steedman, 1978)). Specifically, humans produce more conclusions with an A-C variable order when reasoning in response to a syllogism presented in variable order 1 (A-B, B-C), and they show a pronounced bias in the other direction (that is, they produce more C-A-ordered conclusions) when presented with a syllogism in ordering 2 (B-A, C-B). We aggregate the human and LM responses across all (A-B, B-C) syllogisms and across all (B-A, C-B) syllogisms separately and normalize the aggregated response counts. We find that the PaLM 2 models show a variable ordering effect in the same direction as humans (Figure 4). We compute the magnitude of the efffect for variable ordering 1 by subtracting the mass placed on C-A conclusions from the mass placed on A-C conclusions, $P(\text{C-A}) - P(\text{A-C})$. Similarly, we compute the difference in the opposite direction, $P(\text{A-C}) - P(\text{C-A})$, to estimate the effect magnitude for variable ordering 2. Our results are shown in Figure 4, left), we find that the magnitude of the bias is smallest for the smallest model and increases in larger ones.

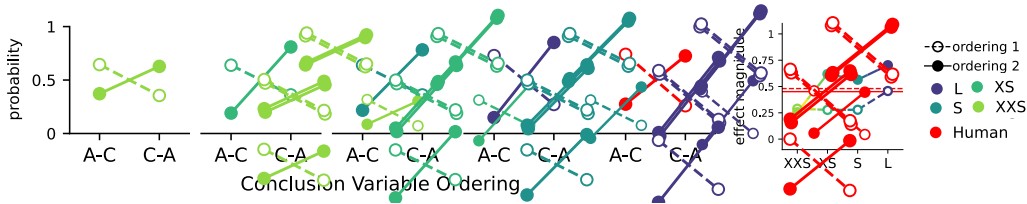

Figure 4: (Left)The marginal probabilities of A-C/C-A ordered conclusion as estimated from human and LM response counts. Humans and LMs show the variable ordering effect. (Right) The magnitude of the variable ordering effect in PaLM 2 increases with model size.

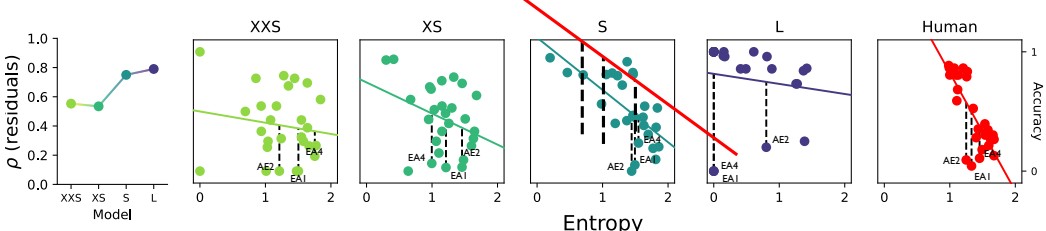

Figure 5: (Right) Each syllogism plotted by accuracy (y-axis) and entropy (x-axis) and the regression line relating the two. Dashed lines black lines show the residuals for each of the top three human syllogistic fallacies. (Left) The result of correlating PaLM 2's residuals with residuals estimated from human data.

**Syllogistic fallacies** In general, humans are well calibrated syllogistic reasoners—their accuracy is inversely correlated with the entropy of their responses (Figure 5; also see Khemlani & Johnson-Laird 2012). In other words, for most syllogisms where humans give incorrect answers, the particular incorrect answers they give vary substantially across individuals and trials. However, there are exceptions to this tendency: in some cases, humans confidently and consistently choose a particular incorrect answer (that is, low entropy coincides with low accuracy). For example, given the syllogism 'no artists are bakers, all bakers are chemists', humans overwhelmingly respond with the logically invalid conclusion 'no artists are chemists'; the correct conclusion, 'some chemists are not artists' is produced only 3% of the time. The distribution over responses elicited from humans for this syllogism has one of the lowest entropies in the Ragni et al. (2019) dataset. We refer to such cases as *syllogistic fallacies* (Newsome & Johnson-Laird, 2006; Khemlani & Johnson-Laird, 2017).

To identify potential fallacies in LMs, we fit a regression line relating entropy (in nats) and accuracy, and then compute the distance from this line (the residual error) for each syllogism. Figure 5 shows the regression lines as well as the top three human syllogistic fallacies, the top three outliers when plotting accuracy against entropy, in humans—we find that these syllogisms are also outliers for the PaLM 2 models (especially so for PaLM 2 Large). Furthermore we correlate the residual errors (for all 27 syllogisms) estimated for the PaLM 2 models with the residual errors estimated for humans and found larger models have stronger correlations (Figure 5, left). Fallacies are a particularly strong test of human-like reasoning performance in this setting: if the models are becoming increasingly human-like, we expect LM accuracy to *decrease* with size on the fallacy syllogisms. We find that LMs do in fact show this trend, even though their overall accuracy increases across the dataset.

**LMs avoid "nothing follows"** An important divergence from human behavior is that LMs rarely conclude "nothing follows", even for the 37 syllogisms for which this is the correct conclusion. Humans also show a reluctance to conclude "nothing follows"; Ragni et al. (2019) analyse this behavior and show that cognitive models struggle to capture this aspect of human syllogistic reasoning. That said, this LM bias goes far beyond a human-like aversion—we observe accuracies around 0% (Figure 6) and very low correlation to human behavior on this part of the dataset. The 'nothing follows' conclusion stands out from the others in that it does not relate A and C, and we find that it is difficult to enable models to generate it in the zero-shot setting (though see Appendix B.3 for a more involved procedures that better elicits that conclusion). We leave futher analysis of this behaviour to future work and beyond this section continue to consider just those syllogisms that derive conclusions other than 'nothing follows.'

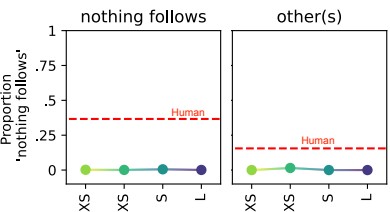

Figure 6: The proportion of 'nothing follows' responses from humans and LMs on (left) the 37 syllogisms whose only valid conclusion is "nothing follows" and (right) the syllogisms that derive conclusions other than "nothing follows".

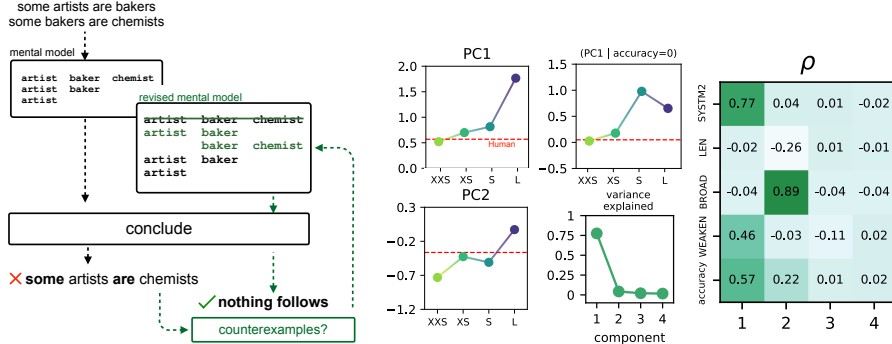

Figure 7: (Left) Schematic of mReasoner deducing an incorrect conclusion before finding conterexamples (system 2 processes shown in green) and updating to the correct conclussion—'nothing follows'. (Center) PC coordinates assigned by projecting PaLM 2's behaviour. (Center - top right) PC1 coordinate assigned to models after setting the probabilities of correct answers to zero. This dimension has 0.66 correlation with **SYSTM2** despite having 0 correlation with accuracy and we find a similar effect of scale in this dimension. (Right) correlation matrix between the coordinates assigned to our mReasoner instantiations and the original parameter values of those instantiations.

## 5 INTERPRETING LANGUAGE MODELS USING MENTAL MODELS THEORY

The Mental Models theory of human logical reasoning (Johnson-Laird, 1983) has been developed over decades to account for a range of human experimental data. The theory takes humans to be resource-limited and simulation-based reasoners (Craik 1967; Lake et al. 2017; Lieder & Griffiths 2019; Johnson-Laird 1983, i.a.), with a potentially high degree of variability. The implementation we use—mReasoner[1](Khemlani & Johnson-Laird, 2022)—captures these aspects of human reasoning with a small set of interpretable hyperparameters that enable it to construct, refine, and draw conclusions from internal 'mental models' of the situations described in a syllogism.

Mental models in mReasoner consist of sets of entities (Figure 7), where an entity is represented by a conjunction of logical properties. For example, Figure 7 (left) illustrates mReasoner constructing a mental model from the syllogism 'some artists are bakers, some bakers are chemists'. Its model (shown in the pane labeled 'mental model') consists of just three enities, the first of whom is an artist who is also a baker and a chemist, the second is an artist and a baker (who may or may not be a chemist, this uncertainty is represented in the Figure with a blank space), and so on. mReasoner constructs and maintains its mental model with a set of actions parameterized by four hyperparameters, which we describe briefly here and in further detail in Appendix D:

- **LEN** ($\lambda \in [1, \infty)$): Determines the average number of entities generated by mReasoner. mReasoner samples a Poisson random variable with mean=**LEN**. The number sampled is the number of entities mReasoner will generate.

- **BROAD** ($\epsilon \in [0, 1]$): Determines the set of individuals the mReasoner samples from. There are two possible sets: a smaller, canonical (biased) set of individuals consistent with the premises (shown in Figure 12) and a broader, complete, set of individuals consistent with the premises. Higher values of **BROAD** indicate that the reasoner is more likely to sample from the complete set.

- **SYSTM2** ($\sigma \in [0, 1]$): The reasoner's propensity to reconsider its conclusion and search for counterexamples. Search is conducted either by adding a new entity to the model, moving a property from one entity to another or by decomposing an entity into two entities (illustrated in Figure 13). Higher **SYSTM2** correspond to a greater liklihood of searching.

- **WEAKEN** ($\omega \in [0, 1]$): Determines the model's reaction to finding a counterexample. mReasoner's options in this case are (1) respond "nothing follows" and (2) avoid "nothing follows", but weaken its response (i.e., amending erroneous *global* conclusions e.g., 'All A are C' to 'weaker' *particular*

---

[1]https://github.com/skhemlani/mReasoner

conclusions e.g., 'Some A are C'). When **WEAKEN** is higher, mReasoner is more likely to weaken and less likely to answer "nothing follows".

We illustrate mReasoner processing the syllogism 'some artists are bakers, some bakers are chemists' in Figure 7. First, mReasoner constructs a mental model (with length governed by **LEN** and content governed by **BROAD**) consisting of the entities discussed above (an artist-baker-chemist, an artist-baker, and an artist). The conclusion 'some artists are chemists' is consistent with this particular model (i.e., the first entity is both an artist and a chemist), but is not true in every model that is consistent with the premises (i.e., the conclusion is not logically valid). If the mReasoner procedure does not trigger a system 2 process, it will (incorrectly) take this conclusion as valid and return it. Alternatively, with probability **SYSTM2** mReasoner will scrutinize the conclusion by amending its model in an attempt to find a counterexample. In this case, mReasoner successfully finds a counterexample by breaking the first entity into two new entities that are still consistent with the premises but which are not consistent with "Some artists are chemists"; in this case, mReasoner corrects its answer to 'nothing follows'.

**Mapping LM predictions onto cognitively meaningful dimensions.** Syllogistic reasoning behaviour is high dimensional; in the set of syllogisms and conclusions we consider, there are 27 syllogisms and eight possible responses to each, for a total of 216. We evaluate mReasoner on each syllogism and represent each instance as a vector in this 216-dimensional space. Finally, we use PCA to identify the top four principal components in this 216-dimensional space. We instantiate 923 mReasoner models, one for each point in a large grid (Table 3 in Appendix).

**Characterizing the space of reasoning behaviours described by mReasoner.** Although mReasoner is characterized by four parameters, we find a single principal component (PC 1) that captures 77% of the variance in the model's behaviour (Figure 7, center). We find that this component loads heavily on **SYSTM2** and, to a lesser degree, on **WEAKEN**. Following Khemlani & Johnson-Laird (2016), we view this dimension as representing *deliberative* reasoning. Similarly, PC 2 loads heavily on **BROAD**. This dimension, however, describes much less behavioural variance in mReasoner. (For the relations between the 4 components and the original parameters, please see Figure 7, right.)

**LMs show signatures of deliberative reasoning.** We project the 216-dimensional vectors describing the human data as well as each of our LMs into this space. This allows us to interpret the LM behaviour, in particular as model size increases, in terms of reasoning strategies (Figure 7). We find that PaLM 2's responses moved upward along PC 1 and PC 2 as models grew larger; in other words, larger LMs show behavioural signatures of deliberative reasoning, their behaviour is more like mReasoner instantiations with high **SYSTM2** and **WEAKEN** values.

**Deliberative reasoning is partly dissociable from accuracy.** PC 1 is strongly correlated with **SYSTM2**, but is also strongly correlated with accuracy. Can the changes in coordinates assigned to PaLM 2 be explained by differences in accuracy alone? To test this, we conduct the same analysis, this time setting the probabilities of the correct answers to 0 for all mReasoner instantiates, LMs, and humans and renormalizing (Figure 7, center). In this control analysis, the accuracy of all models is 0%, but the models still show an increase in deliberative reasoning with size. Here the deliberative component has zero correlation with accuracy but has .6 correlation with **SYSTM2**; correlations with all other parameters are below .15. This result indicates that even the models' errors are more consistent with deliberative reasoning, which provides evidence for nontrivial behavioural signatures of deliberative reasoning as manifested in mReasoner.

## 6 DISCUSSION AND LIMITATIONS

Do language models learn to reason correctly from self-supervised learning alone, even though much of their training data was produced by humans, whose reasoning often deviates from normative logic? We address this question through a detailed examination of PaLM 2's syllogistic reasoning behaviour. We find that (1) the largest LMs make significantly fewer mistakes than humans, but still display systematic errors (Section 4.1), and (2) the mistakes LMs make are only somewhat aligned with human errors, even though LMs are susceptible to several qualitative reasoning biases shown

by humans (Section 4.2). We discuss takeaways, limitations, and connections to broader literature in the remainder of this section.

**Human-like reasoning or accurate reasoning?**   Because of humans' systematic reasoning errors, syllogistic reasoning is a particular clear demonstration of the tension between the two central aims of artificial intelligence: human-likeness and accuracy. We hypothesize that for most applications accuracy is more important; one notable exception is cognitive modeling, where the goal is to better understand human reasoning by developing models that reason like humans.

**Why are LMs more accurate than humans?**   LMs learn from human generated text, which is likely to reflect human beliefs and biases; it is natural to hypothesize that the language modeling objective would incentivize LMs to replicate those biases. We find only partial support for this hypothesis. While the largest model's responses are indeed slightly more correlated with human responses than the smaller ones, for some syllogisms it overcomes human biases and reasons correctly. One possible explanation for this finding is that the data that PaLM 2 models were trained on includes not only natural language text, but also source code (Chowdhery et al., 2022), which may teach models to reason more effectively. The effect of the composition of the LM's training corpus can be tested in a controlled comparison in the future, though retraining state-of-the-art LMs requires substantial computational resources.

**Eliciting LM reasoning**   The space of possible ways to evalute LMs on paradigms from human experiments is fairly large. Evaluations can be done in a zero-shot way, as we did, or in a few-shot way, which may better approximate the training phase used in some reasoning experiments, such as Ragni et al. (2019); see Lampinen (2022). One can generate from the model (Aina & Linzen, 2021), as we did; elicit meta-level judgements (Hu & Levy, 2023; Beguš et al., 2023); or simply compare the probabilities assigned by the LM to possible continuations (Linzen et al., 2016; Dasgupta et al., 2022). Finally, generative approaches can rely on a large set of possible prompts, and can be used with or without "chain-of-thought" incantations whose stated purpose is to cajole the model into revealing its reasoning process. Following preliminary experiments (Appendix B), we focused on zero-shot chain-of-thought; a more systematic evaluation of the different elicitation approaches is an important direction for future work.

**Cognitive science for LM interpretation**   We have used cognitive science to shed light on LM reasoning in two ways. First, we used the biases documented in the cognitive pscyhology as hypotheses for the biases that LMs might acquire. The hypothesis is that since LMs are trained on texts generated by humans, which reflect human biases and beliefs, they will be incentivized to replicate those biases to improve perplexity. This hypothesis is supported by the fact that larger LMs showed stronger human-like biases in some cases (Section 4.2), a phenomenon referred to elsewhere as inverse scaling (McKenzie et al., 2023).

The second, and more novel, way in which we use cognitive science is by using a computational cognitive model based on Mental Models theory to interpret LM behavior. Under the assumption that LM reasoning follows the same heuristic strategies as humans do (Section 5), we can conclude from this analysis that LMs become more deliberative as their size increases. Of course, this is not the only possible mechanism that might underlie LM reasoning. Other accounts of human reasoning have argued that people do in fact apply normative logic rules (Rips, 1994), perform probabilistic inference with constrained resources (Chater & Oaksford, 1999), or combine probabilistic, heuristic and pragmatic reasoning (Tessler et al., 2022); future work can apply our methodology to these theories, which may provide a better explanation of LM reasoning than does Mental Models Theory.

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

## A CONTENT WORDS FOR SYLLOGISMS

Table 2 displays the full list of the content tripples used in our experiments chosen to be minimally semantically associated with each other.

Table 2: The 30 content word triples we use to construct syllogisms (e.g., for the first entry in the table, the variables A, B and C in the syllogism are replaced with *actuaries*, *sculptors* and *writers*, respectively). The words in each triple were chosen to be minimally semantically associated with each other.

| | | |
|---|---|---|
| actuaries, sculptors, writers | assistants, poets, scientists | athletes, assistants, chefs |
| chemists, drivers, dancers | chemists, workers, painters | clerks, butchers, athletes |
| dancers, bankers, riders | doctors, riders, investors | drivers, porters, chemists |
| farmers, surfers, writers | gamblers, cleaners, models | golfers, cyclists, assistants |
| hunters, analysts, swimmers | joggers, actors, carpenters | linguists, cooks, models |
| linguists, skaters, singers | managers, clerks, butchers | miners, tellers, poets |
| models, tailors, florists | nurses, scholars, buyers | planners, sailors, engineers |
| riders, agents, waiters | riders, novelists, linguists | runners, opticians, clerks |
| scientists, novelists, florists | skaters, barbers, cooks | students, cashiers, doctors |
| students, hikers, designers | surfers, painters, porters | therapists, hikers, opticians |

## B  PROMPTING & EVALUATION

Before settling for the generative chain-of-thought evaluation strategy that we focus on in this paper, we explored two additional strategies for eliciting and scoring syllogistic inferences from LMs. First, we explored a multiple-choice approach, where, following the prompt, we computed the mutual information of each of the nine possible conclusions (eight valid conclusions plus "nothing follows"); and second, we explored a simplified binary discrimination approach, where, following the prompt and a particular conclusion, we computed the mutual information of the strings "valid" and "invalid". Of these three methods, chain-of-thought prompting achieved the highest accuracy generally and has stable performance in a wide range of hyperparameters, so we use it in the main text. We do note that the binary discrimination approach has the highest correlation with humans and is the only method that consistently provides the response "nothing follows" when appropriate, and as such is a promising method to explore in future work. The remainder of this appendix provides additional details about the different elicitation methods and the variations on those methods that we explored.

### B.1  ZERO-SHOT CHAIN-OF-THOUGHT

The zero-shot chain-of-thought approach is illustrated in Figure 1. We first describe the inference task: "Choose the conclusion that necessarily follows from the premises or "nothing follows" if none of the other conclusions logically follow, ". We then define the conclusion space, with the string "the possible conclusions are: " followed by the list of all possible conclusions, including "nothing follows". Next, we provide the two premises for the syllogism being queried in the format: 'Premise 1: PREMISE1, Premise 2: PREMISE2, '. Finally, we instruct the LM to produce a reasoning trace by adding "Let's think this through, step by step".

#### B.1.1  ROBUSTNESS TO PROMPT AND DECODING HYPERPARAMETERS

The analyses presented in the main text are based on a decoding process in which we sequentially generate 75 tokens from the LM, with a temperature of 0.5, and take 30 such samples for each combination of syllogism type and content triple. Due to compute limitations we are unable to conduct a systematic exploration of different variations on these hyperparameters, especially for the larger and compute-intensive models; in this section, we focus on PaLM 2 XS. Again due to compute limitations, we draw only 12 samples for each combination of content triple and syllogism pair, as opposed to 30 used in the main text. Finally, with the exception of the analyses investigating the impact of the number of tokens we decode, we set this parameter to 50 (instead of 75 in the main text). As such, the accuracy values we report in this section are not directly comparable to those in the main text. As in the main text, we only report accuracy for the 27 syllogisms that have valid conclusions, and exclude the syllogisms for which "nothing follows" is the correct response.

**Prompts.** In addition to the prompt we used in the main text, which we refer to as `stepxstep`, we consider three variations on the prompt (Figure 9); in all of these experiments, we hold the decoding temperature at 0.5 and the maximum number of decoded tokens at 50:

1. `logically`: The same as `stepxstep`, except the zero-shot reasoning trigger "Let's think this through, step by step" is replaced by "Think logically" (like `stepxstep`, this prompt inspired by a prompt from Kojima et al. 2022).

2. `empty`: This prompt does not include any zero-shot reasoning trigger ("Let's think this through, step by step" is replaced with the empty string).

3. `alt`: We created this prompt in an attempt to mitigate the LMs' reluctace to produce "nothing follows"; here the possibility of a "nothing follows" response is highlighted closer to the end of the prompt and in a more verbose way. This prompt also encourages the model to use the exact wording used in the prompt, and replaces "Let's think this through, step by step" with the slight variation "Let's think step by step".

We report results in Figure 8, the variants show similar behavior though `stepxstep` achieves a modest improvement in accuracy.

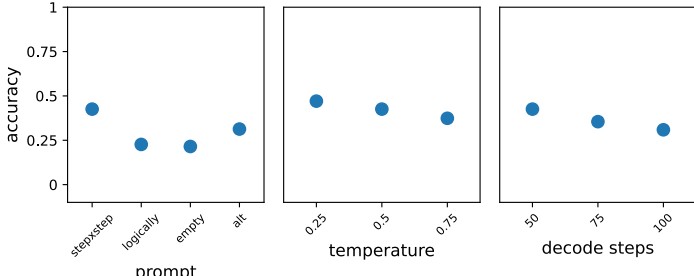

Figure 8: Accuracy for the chain-of-thought prompting method, with different prompts, temperatures and number of decoding steps.

```
stepxstep
Choose the conclusion that necessarily follows from the premises
or "nothing follows" if none of the other conclusions logically
follow,
the possible conclusions are:
"all artists are chemists",
"some artists are chemists",
"no artists are chemists",
"some artists are not chemists",
"all chemists are artists",
"some chemists are artists",
"no chemists are artists",
"some are chemists not artists",
"nothing follows".
Premise 1: all artists are bakers,
Premise 2: some chemists are bakers.
Let's think this through, step by step.
```

```
empty
Choose the conclusion that necessarily follows from the premises
or "nothing follows" if none of the other conclusions logically
follow,
the possible conclusions are:
"all artists are chemists",
"some artists are chemists",
"no artists are chemists",
"some artists are not chemists",
"all chemists are artists",
"some chemists are artists",
"no chemists are artists",
"some are chemists not artists",
"nothing follows".
Premise 1: all artists are bakers,
Premise 2: some chemists are bakers.
```

```
logically
Choose the conclusion that necessarily follows from the premises
or "nothing follows" if none of the other conclusions logically
follow,
the possible conclusions are:
"all artists are chemists",
"some artists are chemists",
"no artists are chemists",
"some artists are not chemists",
"all chemists are artists",
"some chemists are artists",
"no chemists are artists",
"some are chemists not artists",
"nothing follows".
Premise 1: all artists are bakers,
Premise 2: some chemists are bakers.
Think logically.
```

```
alt
Output the conclusion or conclusions that are logically true give
premises.
The possible conclusions are as follows (your output should use th
exact wording): "all artists are chemists",
"some artists are chemists",
"no artists are chemists",
"some artists are not chemists",
"all chemists are artists",
"some chemists are artists",
"no chemists are artists",
"some are chemists not artists",
"nothing follows".
Premise 1: all artists are bakers,
Premise 2: some chemists are bakers.
In some cases, none of these conclusions will be logically valid,
output the words 'nothing follows' in this case.
Let's think step by step.
```

Figure 9: Variations on the prompt we used for the generative elicitation method; the prompt used in the main text is `stepxstep`.

**Decoding hyperparameters** We also vary experiment decoding length and temperature independently (Figure 9). We use the temperatures $\{0.1, 0.25, 0.5, 0.75\}$, holding the decoding length at 50 and using the `stepxstep` prompt. Likewise, we vary the number of tokens decoded between $50, 75, 100$, keeping the temperature at 0.5 and the `stepxstep` prompt. We also find that overall accuracy is largely robust to these ranges.

### B.2 MULTIPLE-CHOICE EVALUATION.

The second approach we present discriminatively ranks each conclusion but otherwise looks very similar to the chain-of-thought approach. We remove the zero-shot chain-of-thought trigger from `stepxstep` and replace it with 'The conclusion that necessarily follows is: ', then feed this to the models and score each of the conclusions. To normalise for idiosyncratic features of each conclusion (i.e., length and prior probability), we use the *mutual information* (Holtzman et al., 2021) between the prompt and the conclusion as the score:

$$\text{MI}(\text{conclusion}; \text{prompt}) = \log P(\text{conclusion}|\text{prompt}) - \log P(\text{conclusion}|\text{""}) \qquad (1)$$

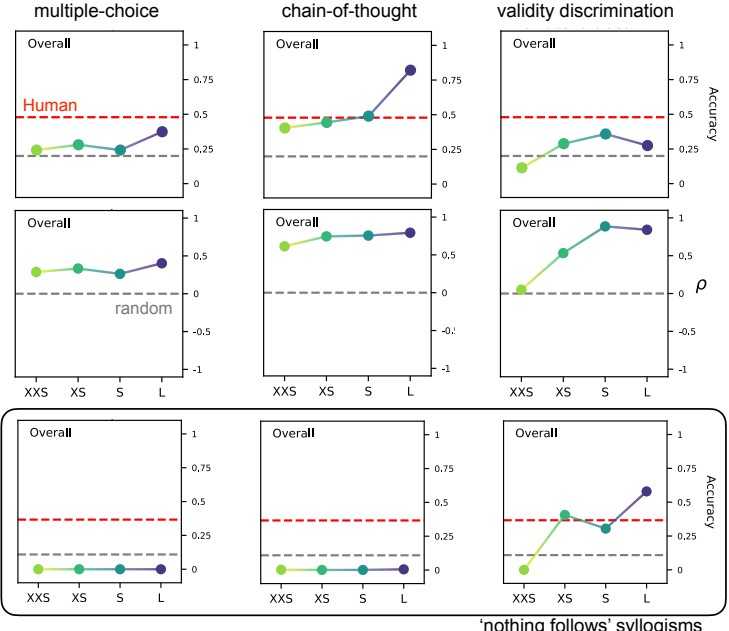

Figure 10: Accuracy across prompting strategies. CoT achieves high accuracy, motivating our choice. Binary validity discrimination yields lower accuracy on the 27 syllogisms we consider in the main text but markedly higher accuracy on the 'nothing follows' syllogisms.' Both outperform 'multiple-choice substantially.

We then renormalise these scores to compute a distribution over the conclusions:

$$P\left(\text{conclusion}_i\right) = \frac{\exp\left(\text{MI}\left(\text{conclusion}_i; \text{prompt}\right)\right)}{\sum_j \exp\left(\text{MI}\left(\text{conclusion}_j; \text{prompt}\right)\right)}, \tag{2}$$

and take the conclusion with max $P\left(\text{conclusion}\right)$ to be the language model's prediction for a given syllogism, content-triple pair.

### B.3 SIMPLIFIED BINARY EVALUATION

While the multiple-choice format allows us to draw a parallel to the human experiments, it poses a significantly harder task than simple binary discrimination. As such, we include a binary discrimination task inspired by Dasgupta et al. (2022). In this setting, we present the model with the prompt "Is this conclusion valid given the premises:" followed by the premises and a single conclusion (we refer to the concatenation of the prompt and conclusion$_i$ as prompt$_i$ below). We do this for all eight possible conclusions (omitting 'nothing follows'). We, again, use the mutual information to score and compute the binary probability of 'valid' as:

$$P\left(\text{`valid'}|\text{conclusion}_i\right) = \frac{\exp\left(\text{MI}\left(\text{`valid'}; \text{prompt}_i\right)\right)}{\exp\left(\text{MI}\left(\text{`valid'}; \text{prompt}_i\right) + \exp\left(\text{MI}\left(\text{`invalid'}; \text{prompt}_i\right)\right)\right)}$$

We compute discrete conclusion decisions by normalizing $P$('valid') for each conclusion into a probability distribution:

$$P\left(\text{conclusion}_i\right) = \frac{P\left(\text{`valid'}|\text{prompt}_i\right)}{\sum_j P\left(\text{`valid'}|\text{prompt}_j\right)}, \tag{3}$$

And taking the conclusion with the largest probability according to Equation 3 to be the model's selected conclusion for a syllogism (the conclusion most likely to be valid according to the model). In this approach, the model's prediction is taken to be 'nothing follows' if $P\left(\text{`valid'}|\text{conclusion}\right)$ does not exceed 50% for any of the conclusions.

## C  FURTHER COMPARISONS WITH HUMANS

This section provides syllogism-level correlations across our dataset.

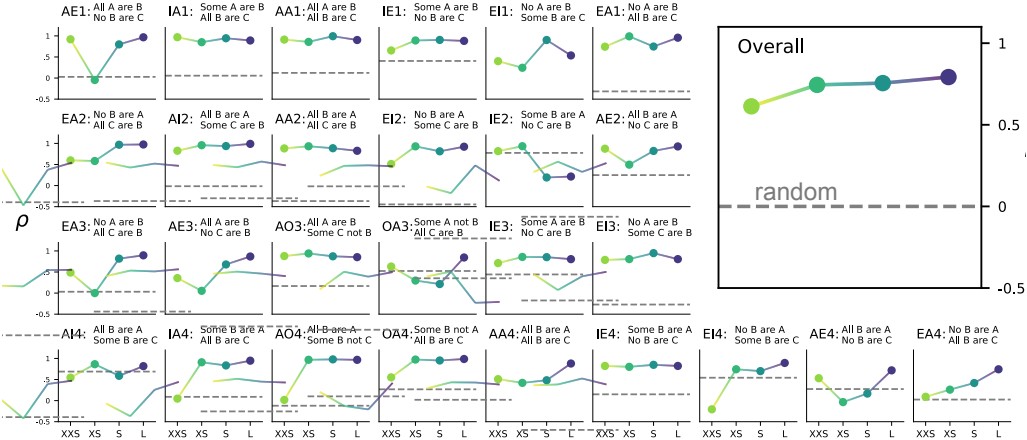

Figure 11: Correlation between language model and probabilities derived from normalizing human responses by syllogism. Syllogisms are partitioned into variable ordering type (by row) and ordered by decreasing human accuracy from left to right. Chance performance (dashed grey line) reflects random guessing. The top right inset shows correlation across the entire dataset.

## D  MREASONER

### D.1  MODEL DETAILS

In this section, we describe the heuristics mReasoner uses to construct its mental models and provide further details on our PCS analysis.

**all** artists **are** bakers
{artist    baker,}

**no** artists **are** bakers
{artist    ¬baker,
 ¬artist   baker }

**some** artists **are** bakers
{artist    baker,
 artist          }

**some** artists **are not** bakers
{artist    ¬baker,
 artist    baker ,
           baker }

Figure 12: The 'canonical sets' used by mReasoner. The canonical set for a syllogism depends on the moods of the syllogism's premises. We show the possible individuals each premise contributes to a syllogism's canonical set here for hypothetic content words 'artists' and 'bakers.'

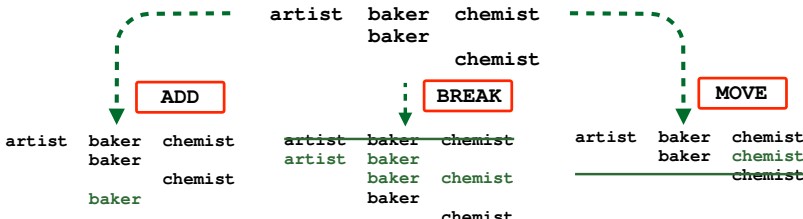

Figure 13: Subroutines used by mReasoner to edit its mental model in order to check for counterexamples. Here denoted as **ADD**, **BREAK**, and **MOVE** following Khemlani & Johnson-Laird (2022). **ADD** adds one more entity to mReasoners mental model. **BREAK** decomposes an entities properties into constituent entities with subsets of those properties. **MOVE** simply moves a property from on entity to another.

## D.2  MREASONER INSTANTIATIONS

We instantiate one mReasoner model for every parameter vector in the grid shown in Table 3. This resulted in a total of 1,296 models. As the models are stochastic, we evaluate each 100 times to estimate their distribution over responses. Due to resource constraints, we discarded models that did not finish these 100 iterations in 60 seconds, leaving us with 923 models spaced relatively evenly over the grid (i.e., this timeout criterion did not favour including some models over others).

Each of the 923 models corresponds to a 216-dimension vector ($27x8 = 216$). We perform PCA on the 923 vectors.

| | | | | | | |
|---:|---|---|---|---|---|---|
| **LEN** | 2.0 | 2.5 | 3.0 | 3.5 | 4.0 | 4.5 |
| **BROAD** | 0.0 | 0.2 | 0.4 | 0.6 | 0.8 | 0.9 |
| **SYSTM2** | 0.0 | 0.2 | 0.4 | 0.6 | 0.8 | 0.9 |
| **WEAKEN** | 0.0 | 0.2 | 0.4 | 0.6 | 0.8 | 0.9 |

Table 3: Parameter grid used to instantiate our mReasoner models.

