# OpenReview forum: "A Systematic Comparison of Syllogistic Reasoning in Humans and Language Models"
_ICLR.cc/2024/Conference — ICLR 2024 Conference Withdrawn Submission_

### Official Review · Reviewer_Q1gw · 2023-10-30

**Soundness:** 1 poor
**Presentation:** 4 excellent
**Contribution:** 3 good
**Rating:** 3
**Confidence:** 4

**Summary:**

The paper focuses on a very important issue in human logical reasoning, namely syllogisms, in which reasonings are done based on two premises. This paper conducted a system comparison between such an ability between LLMs in the PaLM 2 family and humans.

**Strengths:**

1. The subject matter is very interesting and very well-grounded in theories in cognitive science.
2. The whole systematic investigation is very well-designed.

**Weaknesses:**

My only concern is that it is quite unclear how the human data was post-processed as each syllogism is associated with 139 responses. If there is no further post-processing as the one on LM-produced data, then this is an insurmountable issue for me. I am saying this because the LM's outputs are actually majority voting results based on 30 runs. If humans did not do the same, then, for me, machines and humans were doing two very different tasks, making the conclusions extremely biased and, therefore, unreliable. Thinking of a situation where one asks an LM to run 30 times on a test case and 20 of them are correct, the accuracy is 100% after majority voting. Nevertheless,  if one asks humans to do the same but without majority voting, then the accuracy is only 66%, but their (i.e., humans and the LM) behaviours are very similar to each other.

For me, a better solution is dropping the majority voting and computing distributional similarities when comparing LMs with humans (I mean directly rather than the one in section 4.2).

**Questions:**

I have no further questions.

---

> ### Author Response · Authors · 2023-11-16
>
> Thank you for your review and constructive feedback. We will respond together to both of the weaknesses you identify. In general, comparing computational models to human laboratory studies is an active area of research (see Section 6, paragraph 4 for a discussion of this literature). Just for the syllogism task, dozens of computational models have been proposed, as well as numerous methods to link their output to human behavior (see Ragni et al. 2019, who scrutinize the linking functions implicit in cognitive models of this domain and find that even model designed explicitly for syllogistic reasoning can be linked to behaviour in myriad ways that have meaningful effects on behaviour). And as we mentioned in other responses to reviewers hrcw and Wr2g, there are many degrees of freedom in the evaluation paradigm: the field hasn’t yet converged on a set of best practices for evaluating LLM reasoning, and there are multiple possible decision points, such as the choice between generative prompt-based and scoring-based methods, the particular prompt and decoding hyperparameters, and so on. Appendix B extensively details our exploration of many of these options, with significant computational cost, but we certainly agree that there are even more options that one could explore, though a single paper may not be able to explore all of them.
>
> As for the specific issue you mention: one particular issue with comparing LMs to human data is that in practice we have only one LM, but human studies include many different human participants. Following your example, if we ask a single human participant to respond to the same problem 30 times, we expect significant consistency among their reactions, whereas for an LM’s response will vary based on decoding hyperparameters (e.g. temperature). The method we focus on in the main paper – repeated samples generated from a chain-of-thought prompt – is motivated by an understanding of humans as arriving at syllogistic conclusions by (1) computing a distribution over responses internally, and (2) converting this distribution into a discrete decision by returning the most likely member of the support.
>
> Finally, we note that because we have 30 different content word triples for each syllogism type, we do obtain distributions across possible responses even for the prompt-based method, which we compare to the distributions over the set of conclusions produced by humans for each syllogism. In the methods based on the LMs’ logprobs, which we explore in Appendix B, variance in the distribution arises both from the variance across items and from the probabilistic nature of the LMs’ predictions. Again, given the huge space of possible evaluation methods, we did not also include a version of the prompt-based method that incorporated both the variance across content triple and the variance across multiple generation chains for the same item into the distribution; however, we plan to add this version, as you propose, to the next version of the manuscript.
>
> Citations
> Ragni, M., Dames, H., Brand, D., & Riesterer, N. (2019). When does a reasoner respond: Nothing follows? CogSci, 2640–2546.

---

> > ### Comment · Reviewer_Q1gw · 2023-11-17
> > **Response to the rebuttal**
> >
> > Thanks for your response.
> >
> > > if we ask a single human participant to respond to the same problem 30 times
> >
> > I was not expecting an experiment as such, but something like a majority voting on the human responses is doable.
> >
> > > The method we focus on in the main paper – repeated samples generated from a chain-of-thought prompt – is motivated by an understanding of humans as arriving at syllogistic conclusions
> >
> > But you didn't use all those 30 samples. Instead, you took the conclusion that was produced most frequently
> > across the 30 samples. If you directly compare the resulting conclusion to human responses, then it is clearly unfair.
> >
> > > Finally, we note that because we have 30 different content word triples for each syllogism type, we do obtain distributions across possible responses even for the prompt-based method, which we compare to the distributions over the set of conclusions produced by humans for each syllogism.
> >
> > I see that you also include an experiment (which is in the appendix) that compares the distributions but since the main comparisons do not seem fair, the reliability of the conclusions is questioned.

---

### Official Review · Reviewer_Wr2g · 2023-10-31

**Soundness:** 4 excellent
**Presentation:** 3 good
**Contribution:** 2 fair
**Rating:** 5
**Confidence:** 3

**Summary:**

The paper analyzes the behavior of PaLM 2 on syllogisms and concludes that (1) it is generally more accurate than humans (and more so with a bigger size), (2) its judgment is still correlated with human judgment, and (3) it also has biases in incorrect logical reasoning (more so with a bigger size).

**Strengths:**

The paper makes a clear contribution in understanding how humanlike LMs are by exploiting the already existent study of human performance on syllogisms. This study allows us to make a fine-grained examination of 64 syllogism types (2 premises * 4 orderings * 8 moods), each of which has human performance recorded from 139 participants. This approach makes the claims made in the submission believable.

**Weaknesses:**

- The findings are useful but not surprising.
- Some natural settings are omitted, including few-shot prompting which is only mentioned. While the paper can't do everything, it seems a bit incomplete to not check the possibility that few-shot (which is more aligned with the human data) makes a big difference.
- Only the PaLM 2 model family is considered, though that's understandable from a limited budget point of view. But we may not be able to discount the possibility that the LM behavior changes significantly with, e.g., GPT-4.

**Questions:**

See above.

---

> ### Author Response · Authors · 2023-11-16
>
> Thank you for your review and constructive feedback. We provide detailed replies to the listed weaknesses and questions below:
> 1. Our perspective is different: we think our results are both useful and surprising! We do not think it is a priori obvious that LMs would, as we found, recapitulate subtle biases in human reasoning, such as the specific pattern of sensitivity to the ordering of variables in a syllogism, while at the same time also achieving superhuman accuracy across the board, and displaying highly un-humanlike behaviours on some parts of the datasets (e.g. the syllogisms for which ‘nothing follows’ is the valid conclusion). This complex pattern of results is unexpected, in our view, and opens up multiple directions for future research.
> 2. We agree: while we spent a significant amount of computational resources exploring an extensive range of ways to elicit logical reasoning from LLMs (see Appendix B), there are certainly even more dimensions in this space that one would ideally explore. One of these dimensions is few-shot reasoning, as the reviewer points out. We have been working to further extend the space of possible reasoning methods, in particular in the few-shot direction, and will include those results in an updated version of the paper.
> 3. As we mentioned in our responses above, the hypothesis we test is that human reasoning biases reflected in the models’ training data will affect models’ reasoning, and as such we are limited to LLMs that are just pretrained, without additional human intervention through RLHF or supervised fine-tuning; most of OpenAI’s recent models, such as the GPT-4 model mentioned by the reviewer, do not have this property. That being said, since submitting the paper we have been extending our experiments to the Llama model family, as we have mentioned in our response to another reviewer. We expect to have results from the Llama experiments within the next few weeks, and we will incorporate them into the next version of the paper.

---

### Official Review · Reviewer_acDo · 2023-10-31

**Soundness:** 2 fair
**Presentation:** 3 good
**Contribution:** 2 fair
**Rating:** 3
**Confidence:** 4

**Summary:**

This paper focuses on comparing the logical reasoning abilities on syllogism between language models (PaLM 2 family) and human. The authors conduct tests on language models using 64 syllogistic forms about 30 content words, and compare their performance to the result from human reasoning in cognitive psychology research. Furthermore, they use a computational cognitive model based on Mental Models theory to explain and validate the logical reasoning of language models.

**Strengths:**

1. This work covers various human biases in syllogistic reasoning, presenting an in-depth comparison through extensive experiments. It examines the distribution of responses, variable ordering, and generation of syllogistic fallacies in both language models and humans.
2. By employing the Mental Models Theory, this work provides a fresh perspective on understanding the logical reasoning abilities of language models.

**Weaknesses:**

1. The deductive reasoning capability of LLMs has been widely explored. The comparison of reasoning abilities between humans and language models in syllogism has already been explored in Dasgupta et al (2022). And Saparov & He (2022) also utilizes similar controlled techniques for analyzing the capabilities of language models. Besides, this work directly utilizes human results on syllogistic reasoning from previous work and only explore PaLM 2 models on a small set of data, which limits the contribution of this paper.
2. The motivation to use mReasoner and the explanation of its details are not well presented.
3. The description of the datasets and models for experiments are insufficient, such as the size of model parameters. Conducting experiments on different language models would enhance the conclusion of the study.
4. There are some errors:
> - In Section 2.3, Saparov & He (2022) and Saparov et al. (2023) both analyze natural language, contrary to what the authors claim about formal logic forms.
> - There are spelling typos, such as "efffect" in the third paragraph of Section 4.2.

**Questions:**

1. This paper only focuses on syllogisms about 30 content words, which are all about person characters/professions. This limits the reliability of experimental observations and conclusions. Compared to other domains, syllogisms on person characters seems to have less content effects problems?
2. The paper mentioned that "we will use the less confusing term “variable ordering" in Section 2.1, how did you select less confusing terms?

---

> ### Author Response · Authors · 2023-11-16
>
> Thank you for your review and constructive feedback. We provide detailed replies to the listed weaknesses and questions below:
>
> Weaknesses
> 1. Thanks for asking us to clarify the difference between our work from prior work. You are absolutely right that our paper is not the first one to compare LLM and human reasoning; in particular, as we acknowledge in the paper, our work is in many ways an extension of Dasgupta et al. (2022). Reasoning is an immensely complex phenomenon, however. Dasgupta and colleagues focused on a specific aspect of this phenomenon: “content effects”, or the bias to prefer conclusions that are plausible in the real world, independently of whether they follow from the premises. By contrast, we focus on an orthogonal dimension of reasoning: logical fallacies and biases that have to do with the formal structure of the syllogism, as opposed to its content. As such, we design our materials to be as neutral as possible such that all conclusions are equally plausible. In general, because logical reasoning is such a complex phenomenon, we do not believe that it can be completely characterized by a single conference paper; in fact, we anticipate that over the next few years, comparisons between human and machine cognition will be a growing field.
> As for Saparov & He (2022): as you mentioned correctly, Saparov and He evaluate logical reasoning in proofs that are generated from templates, in a controlled setting; in this sense, our methodology is similar to theirs. But there is a crucial difference: unlike Saparov & He, the focus of our paper is on comparing the models’ behavior to human reasoning biases. Much like the space of human-LLM comparisons we discussed in the previous paragraph, we believe that the space of controlled investigations of LLM reasoning is quite large, and can be fruitfully explored in more than one paper. In the revised version of the manuscript, we will sharpen the discussion of the relationship between our paper and related work.
>
> 2. Complementing the descriptive behavioral results reported in section 4, we use mReasoner – a computational cognitive model designed to shed light on human reasoning behavior – to provide intuition about the underlying factors that give rise to LLMs’ reasoning errors. Here, the methodological bet is that the heuristic reasoning processes proposed by Mental Models Theory, the predominant theory of human syllogistic reasoning, may correspond to the LLMs’ reasoning processes. The basic idea of this cognitive theory is that humans reason by instantiating mental models that have the properties defined by the syllogisms, but due to resource constraints, humans often instantiate only some of the possible models, which can lead them astray when reasoning about these syllogisms. We find that larger language models engage in processing that in humans has been characterized as “deliberative”; this is reflected both in the models’ overall accuracy and in their error patterns. From a methodological point of view, we see the experiments described in this section as a test case in how to properly leverage the long history of model building in cognitive science to advance LLM interpretability. We will make our motivation for using this cognitive model clearer in the revised version of the manuscript.
>
> 3. We agree that it would be helpful to include more details about the LLMs we tested; unfortunately, the state of affairs in LLM research these days is such that even basic details such as the number of parameters in most LLMs are not publicly available (this is the case for both Google and OpenAI models). We also agree that it would be great to test as large a variety of language models as possible; as we mentioned to Reviewer hrcw, we do note that the hypothesis we test is that human reasoning biases reflected in the models’ training data will affect models’ reasoning, and as such we are limited to LLMs that are just pretrained, without additional human intervention through RLHF or supervised fine-tuning. This limits the number of models we can test. That being said, since submitting the paper we have been extending our experiments to the Llama model family, and we will likely have results within the next few weeks, which we plan to incorporate into the next version of the paper.
>
> 4. Thank you for pointing out these exposition errors and typos; we will address them in the next draft.

---

> > ### Author Response · Authors · 2023-11-16
> >
> > Questions:
> > 1. That’s correct - the content words we used are all “bland” profession names, such as actuaries, sculptors, and writers. As such, we do not expect to observe content effects. But this is a feature, not a bug! Our goal in this paper was to control for content effects (effects which, as mentioned above, were studied extensively in Dasgupta and colleagues’ work), and focus instead on the formal, logical aspects of syllogistic reasoning, where a comparison between humans and LLMs have not been reported yet. In preliminary experiments that were not included in our original submission, we found that the models’ reasoning follows similar patterns even when the profession names are replaced with truly content-less ‘nonsense’ terms (e.g ‘all spluct are zufd’). We will update the paper with the results of these experiments and will explain our motivation for using semantically “bland” content words more clearly.
> >
> > 2. In the traditional logic and psychology literature, the technical term for the order of variables across the premises of a syllogism is the syllogism’s “figure”. As we do not expect our audience to be familiar with this term, and to avoid confusion with the sense of the word “figure” that refers to plots of the data, we chose to replace this ambiguous term with the 'layman’s' term “variable orderings”.

---

### Official Review · Reviewer_hrcw · 2023-11-06

**Soundness:** 3 good
**Presentation:** 3 good
**Contribution:** 3 good
**Rating:** 6
**Confidence:** 3

**Summary:**

This paper reports results on an family of LLMs' (the PaLM 2 models) zero-shot performance on syllogistic reasoning and compares it to experimental data from human participants.  Such inferences have lay at the heart of the field of psychology of reasoning for decades, and so this paper attempts to leverage that literature for understanding LMs.  They find that model performance increases with scale, and that (even in the case of "super-human performance") many of the errors the models make are similar to the ones that humans do.  The authors further compare the models' responses to a cognitive model of reasoning that was fit to human data, to identify which factors are responsible for the change in behavior with scale.

**Strengths:**

- Uses cognitive science experiments and methods to understand large language models and make relatively direct comparison with human behavior.
- Detailed analysis of model responses, e.g. looking at variable-ordering effects, beyond just summary statistics like accuracy.
- Compares LM responses to a cognitive model fit to human data, to identify which factors drive model behavior at scale.
- Appendix compares different prompting strategies as well.

**Weaknesses:**

- Results are only presented on one family of models, which are behind a closed API.  This has two problems: (1) While it is nice that this family has four scale variants, the results are still presented as about "LMs" in general, whereas we have learned about one in particular.   Another model family or two would help understand whether the results are in fact general. (2) Behind a closed API, the results are not necessarily reproducible.  I'd like to see at least one open model as well.
- The dataset used for evaluation is relatively small.
- Difficult choices are made for parsing LM generation to decide when it has generated a conclusion to the syllogism.  I would have liked to hear more about this choice point and whether they looked at other formats (e.g. multiple choice).
- The tendency of the model to _never_ output "nothing follows" means that all of the interesting analyses in the paper were on a subset of the syllogisms (27/64) where nothing follows is never correct.  This also limits the generality of the results.

**Questions:**

- Up to 4 conclusions are valid: how then, was accuracy measured?  If the output is one of the valid conclusions?  And is this the same as in the human experiment?
- Figure 5: the correlation between entropy and accuracy seems to "go away" for the Large model; do you have any thoughts on why this is?
- It was a bit opaque to me how the mReasoner model was fit to the human data.  Can you say more about this process?  And, more importantly, is it a good fit to the data?

---

> ### Author Response · Authors · 2023-11-16
>
> Thank you for your review and constructive feedback. We provide detailed replies to the listed weaknesses and questions below:
>
> Weaknesses
> 1. We agree that it would be great to test as large a variety of language models as possible. We do note that the hypothesis we test is that human reasoning biases reflected in the models’ training data will affect models’ reasoning, and as such we are limited to LLMs that are just pretrained, without additional human intervention through RLHF or supervised fine-tuning. This limits the number of models we can test. That being said, since submitting the paper we have been extending our experiments to the LLaMA(-2) model family, which is a set of open LLMs, and we will incorporate the results of these experiments into the next version of the paper.
>
> 2. Regarding the size of the dataset: Our strategy in this project is to perform a detailed comparison between humans and LLMs on a specific, well-defined phenomenon for which an extensive understanding of human behavior is available in the cognitive psychology literature.  As such, our dataset includes 1920 syllogisms, 30 instances each of all 64 possible Aristotelian syllogism types. We’re not sure that this dataset is particularly small, but in either case, these 1920 instances give us exhaustive coverage of this phenomenon. While we agree with the reviewer that extending the range of reasoning challenges would be valuable in future research, we believe that our detailed analysis of fundamental syllogistic reasoning already contributes a rich pattern of results to the literature.
>
> 3. We agree that there are many degrees of freedom in the evaluation paradigm: the field hasn’t yet converged on a set of best practices for evaluating LLM reasoning, and there are multiple possible decision points, such as the choice between generative prompt-based and scoring-based methods, the particular prompt and decoding hyperparameters, and so on. App. B extensively details our exploration of some of these options. Overall, the conclusion of this appendix is that prompt-based methods provide the best accuracy, and the qualitative results are robust to the choice of prompt and decoding methods we were able to explore. This motivates our decision to use this method in the main text.
>
> 4. Regarding the models’ reluctance to output “nothing follows” responses when prompted with instructions: we were also surprised by this behavior! All the more reason, in our view, to document it and pose this issue as a challenge for future language models. In App. B.3 we describe an evaluation method that we refer to as “validity discrimination”; this method can be used to implicitly derive “nothing follows” responses with greater frequency than the prompt-based method. The tradeoff is that the accuracy of this method is quite a bit lower than that of the prompt-based method. In the main text of the paper, we chose to side with the best-performing model, focusing on a subset of the syllogisms as the best compromise.
>
> Questions:
> 1. Thanks for this question. Indeed, for some syllogisms, more than one conclusion is valid. When that is the case, we compute accuracy as the proportion of responses that include any of the valid conclusions. This does mean that for some syllogisms chance accuracy (random guessing) is higher than for others (see the dashed gray line in Fig 2). We appreciate that this point could be made more clearly in the paper, and will address this in the next version of the manuscript.
>
> 2. Regarding the correlation between entropy and accuracy in Fig 5: in fact, the largest model still displays a large correlation between entropy and accuracy, if we remove three stark outliers that cause the regression line to be much flatter than for other models and humans (i.e. the model shows an inflated fallacy effect; similarly the largest model shows the, numerically, largest figural effect; Fig 4, right), the coefficient of the correlation between entropy and accuracy for the largest model is -0.62 when the top three outliers are removed, compared to -0.12 when the outliers are included. We appreciate your mentioning this as it is theoretically important to the aims of the paper (the LMs often show inflated biases), and we will include an explanation and additional analyses in the next version.
>
> 3. Thanks for your questions about the connections between the mReasoner cognitive model and the human data. In fact, we do not directly fit mReasoner instantiations to human data (or language model data); instead, we create 1,296 instances of the mReasoner model, one for each vector of hyperparameters in a large grid (App. D.2), which follows the same protocol as in [1] and compare PaLM2 against the behavior of the resulting models with a novel method. We will work to clarify this point further in the paper.
>
> Citations:
> [1] Khemlani, S., & Johnson-Laird, P. N. 2016. How people differ in syllogistic reasoning